# Prediction of ciprofloxacin resistance in hospitalized patients using machine learning

Igor Mintz [1,2], Michal Chowers[3,4] & Uri Obolski [1,2 ✉]

## Abstract

**Background** Ciprofloxacin is a widely used antibiotic that has lost efficiency due to extensive resistance. We developed machine learning (ML) models that predict the probability of ciprofloxacin resistance in hospitalized patients.

**Methods** Data were collected from electronic records of hospitalized patients with positive bacterial cultures, during 2016-2019. Susceptibility results to ciprofloxacin ($n = 10,053$ cultures) were obtained for *Escherichia coli, Klebsiella pneumoniae, Morganella morganii, Pseudomonas aeruginosa, Proteus mirabilis* and *Staphylococcus aureus*. An ensemble model, combining several base models, was developed to predict ciprofloxacin resistant cultures, either with (gnostic) or without (agnostic) information on the infecting bacterial species.

**Results** The ensemble models' predictions are well-calibrated, and yield ROC-AUCs (area under the receiver operating characteristic curve) of 0.737 (95%CI 0.715–0.758) and 0.837 (95%CI 0.821–0.854) on independent test-sets for the agnostic and gnostic datasets, respectively. Shapley additive explanations analysis identifies that influential variables are related to resistance of previous infections, where patients arrived from (hospital, nursing home, etc.), and recent resistance frequencies in the hospital. A decision curve analysis reveals that implementing our models can be beneficial in a wide range of cost-benefits considerations of ciprofloxacin administration.

**Conclusions** This study develops ML models to predict ciprofloxacin resistance in hospitalized patients. The models achieve high predictive ability, are well calibrated, have substantial net-benefit across a wide range of conditions, and rely on predictors consistent with the literature. This is a further step on the way to inclusion of ML decision support systems into clinical practice.

### Plain language summary

Ciprofloxacin is an antibiotic commonly used to treat various infections. Due to the frequent use of ciprofloxacin, bacteria have developed high rates of resistance to it, which means they continue to grow, reducing the effectiveness of treatment. The aim of this study was to develop computer code to predict ciprofloxacin resistance in hospitalized patients. We used data from medical records and tests of whether particular bacteria could be killed by antibiotics from a large hospital in Israel to develop the computer code. The computational model accurately predicted resistance. This model could enable antibiotic treatment to be more appropriately targeted to patients that would benefit from it and reduce the amount of bacteria resistant to ciprofloxacin.

[1] School of Public Health, Tel Aviv University, Tel Aviv, Israel. [2] Porter School of the Environment and Earth Sciences, Tel Aviv University, Tel Aviv, Israel. [3] Meir Medical Center, Kfar Saba, Israel. [4] Sackler School of Medicine, Tel Aviv University, Tel Aviv, Israel. ✉email: uriobols@tauex.tau.ac.il

Antimicrobial resistance (AMR) has developed into a global public health crisis. AMR often emerges rapidly in bacterial populations, and the effectiveness of newly introduced antibiotics can substantially drop after a few years of clinical use[1,2]. In settings of high resistance levels, such as treatment of hospitalized patients, it may become challenging to find empiric antibiotic treatments which will be effective, while minimizing collateral resistance[3]. Such inappropriate empirical treatment is associated with the prevalence of AMR[4]. Despite guidelines[5], literature on collateral damage of antibiotics[5,6], and stewardship initiatives[7], the frequency of bug-drug mismatch in empiric treatment often remains high[4,8].

A notable example of a broadly used antibiotic, with increasing concerns about its resistance frequencies, is ciprofloxacin. Ciprofloxacin is a fluoroquinolone antibiotic, which has been widely used since the early 2000s and is currently on the World Health Organization's List of Essential Medicines[9]. Ciprofloxacin is effective against various gram-negative bacteria, and to a lesser extent gram-positive bacteria, and is used in the treatment of urinary tract, respiratory tract, bone and joint, intra-abdominal, and other infections[10,11]. Hence, ciprofloxacin has been the drug of choice for many infections both in in- and out-patient settings. High consumption rates over decades inevitably increased resistance to the drug[12–14], with an additional indirect effect on non-consumers[15], impeding effective therapy[16]. However, reversion to high levels of sensitivity to quinolones is rapid upon decrease in quinolone consumption[17]. Therefore, minimizing unnecessary ciprofloxacin use can have substantial public health impact.

The use of machine learning (ML) in the context of AMR has been rapidly increasing with the availability of electronic medical records (EMRs) and development of new algorithms. ML models are potentially nearing the point where they can support clinicians' decisions of empiric therapy, by providing rapid predictions of resistance[18,19]. Hence, constant improvement of the methodology and outcomes of such models is of high importance. In the context of ciprofloxacin, prediction models have been scarce and limited to community-acquired urinary tract infections[20], only to intensive care units[21], specific site of infection[22], or to specific subsets of patients[23].

In this study, we developed an ensemble ML model that predicts resistance to ciprofloxacin based on hospitalized patients' EMRs. Importantly, we include as variables relevant frequencies of resistance within the hospital, and not solely the examined patient's EMR. Our models are applied to two settings: assuming that the infecting bacterial species is unknown (a bacteria-agnostic dataset) or known (the bacteria-gnostic dataset), with resulting test-set AUC values of 0.737 (95%CI 0.715–0.758) and 0.837 (95%CI 0.821–0.854).

Furthermore, explainability methods are used to analyze important predictors of resistance in our ML models.

## Methods

**Data.** Data were retrieved from Meir Medical Center, a hospital in Israel which serves approximately 600,000 residents. EMRs of patients who had positive bacterial cultures that were tested for ciprofloxacin susceptibility between the years 2016-2019 were retrieved. The data contained information regarding patients' demographics, functional status, previous antibiotics usage and previous hospitalization within the previous year, bacterial pathogen, and susceptibility results. For gram-negative bacteria in urine or wound culture, VITEK 2 (bioMerieux, Durham, NC) was used. For all isolates from blood or for gram-positive bacteria, in urine, wounds, or blood cultures, disk diffusion with CLSI breakpoints was used. Bacterial cultures demonstrating intermediate resistance results were regarded as resistant.

Additional features related to previous infections with resistant bacteria, previous antibiotic usage, and previous hospitalizations were engineered from the patients' EMRs. The final dataset contained 10,053 susceptibility test results of 5540 patients and 73 variables (see Supplementary Data 1). These data were used to create two data sets: bacteria gnostic (the whole data) and bacteria agnostic (without 20 features related to the bacteria). The train-test split was performed based on calendar time, rather than randomly. This minimizes chances of "data-leakage", where training on future observations holds information on past observations. Furthermore, such a split emulates a real-world scenario where the model can be trained up to a certain point and then used in the clinic from that point onwards, and is considered a form of external validation[24–27]. Each dataset was divided into a training set (75% of all samples) and a test set (25% of all samples), based on the date the culture was taken (Fig. 1). These datasets are mutually exclusive - all the presented results were obtained when training the models solely on the training set, and testing them on the independent test set.

**Machine learning algorithms.** We used an ensemble of several ML algorithms, which we term 'base learners': LASSO penalized logistic regression[28], random forest[29], gradient-boosted trees[29], and neural networks[29]. The base learners' hyperparameters were optimized using 200 random searches[30] with a five-fold, time series cross-validation. To improve the predictions of the four base learners, a stacking technique was applied. In this technique, the predictions of the base learners are given as inputs to a second-level learning algorithm (super learner). The super learner was a logistic regression algorithm trained to optimize the predictions[31]. We adopted a process described elsewhere[32] to train the super learner on time series data (Figure S1 in the Supplementary Material). This resulted in a single ensemble model whose output is the predicted probability of the culture result to have resistance to ciprofloxacin. The tuned hyperparameters are shown in Supplementary Data 2. Model performance was evaluated using the area under the receiver operating characteristic curve (ROC-AUC) metric. Confidence intervals (CI) were calculated using 5,000 bootstrap samples of the test-set data. Model agnostic approximation of the Shapley additive explanations (SHAP) was performed with "kernel SHAP"[33], employing 300 background samples from the training data and calculating the SHAP values of the entire test set.

**Decision curve analysis.** A decision (also known as a utility) curve analysis, which is increasingly recognized as valuable in clinical predictive modeling[26], was performed using the predictions of our ensemble model on the test-set. A decision curve is a graphical representation of the trade-offs between the benefits and costs of a particular treatment or intervention, when administered according to a prognostic algorithm. It is used to evaluate the overall utility of the algorithm by considering both the magnitude of the benefits and costs of no-treatment and redundant treatment, and the likelihood of these results based on prevalence of the outcome and the algorithms' prediction abilities. In such an analysis, the standardized net benefit (sNB) of a decision is defined by the following equation:[34,35]

$$sNB = TPR - FPR \frac{1 - f_{res}}{f_{res}} \frac{p_t}{1 - p_t} \qquad (1)$$

where TPR and FPR are the true- and false-positive rates, respectively; $p_t$ is a threshold probability; and $f_{res}$ is the frequency of resistant infections. In our case, $p_t$ is the threshold probability above which a decision maker (i.e., clinician) is willing to act as if the infection is resistant to ciprofloxacin. This implies that the cost of falsely deciding that an infection is susceptible to ciprofloxacin is $p_t/(1 - p_t)$ fold the benefit of correctly deciding it is

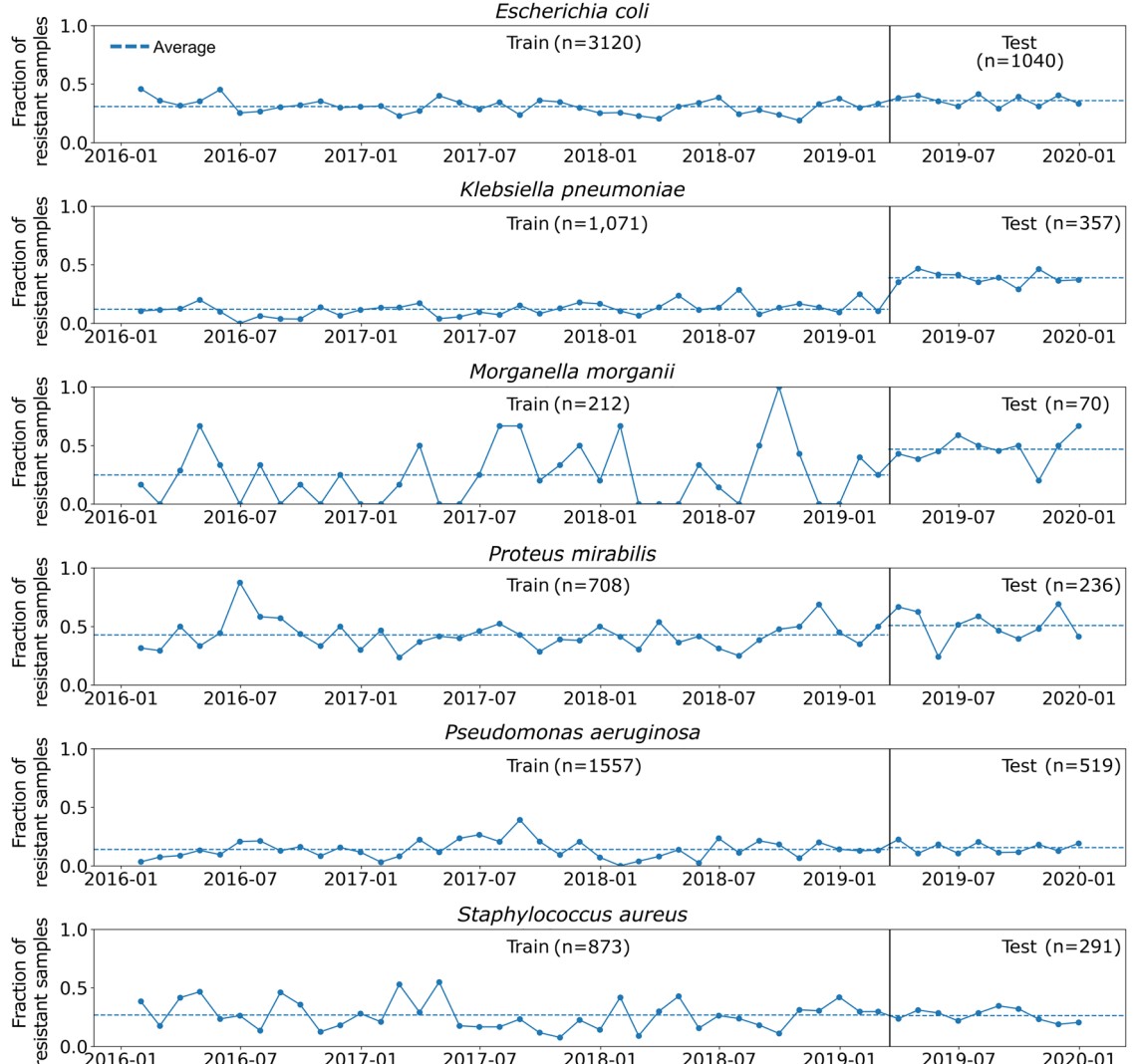

**Fig. 1 Ciprofloxacin resistance time-trends stratified by bacterial species.** Points connected by solid lines are the average monthly ciprofloxacin resistance frequencies. The dotted horizontal lines represent the average resistance in the training and test sets, which are separated by the black vertical lines.

susceptible to ciprofloxacin. Hence $p_t/(1 - p_t)$ is also termed the cost-benefit ratio. For example, assume clinicians will not treat an infection with ciprofloxacin when they know that the probability of ciprofloxacin resistance is above 0.2, but will treat them with ciprofloxacin otherwise. The clinicians are hence implicitly willing to inefficiently treat one patient with a ciprofloxacin resistant infection for every four patients with susceptible infections, yielding a cost-benefit ratio of 1:4.

The sNB of the model merges all the above-mentioned parameters into a single number for each threshold, and hence produces a curve. This curve is compared to two simple decision strategies: assuming that every infection is resistant (all resistant) and that no infection is resistant (all susceptible). The sNB can reach a maximum value of 1, equivalent to assuming that all resistant and susceptible cases are treated correctly (TPR = 1 and FPR = 0).

Analyses were performed with Python 3.7[36], using the following packages: Numpy 1.20.3[37], Pandas 1.3.5[38] and Scikit-learn 1.0.1[39] for data processing; Scikit-learn, XGBoost 1.5.0[40], and Tensorflow 2.4.1[41] for modeling; Matplotlib 3.5.0[42] for plotting; and SHAP 0.40.0[43] for variable influence.

**Ethics approval.** The study was approved by the Institutional Review Board (Helsinki) Committee of Meir Medical Center.

Since this was a retrospective study, using archived medical records, an exemption from informed consent was granted by the Helsinki Committee.

**Reporting summary.** Further information on research design is available in the Nature Portfolio Reporting Summary linked to this article.

## Results

We trained four base learners, and an ensemble model composed of these base learners, to predict ciprofloxacin resistance for six bacterial species. The demographics and basic clinical characteristics corresponding to the cultures' patients are shown in Supplementary Data 3. We note that *K.pneumoniae* and *M.morganii* had a higher proportion of resistant samples in the test set, which potentially may harm predictions. Regardless, our algorithms were able to generalize successfully and achieve high ROC-AUC scores.

ROC-AUC scores and calibration plots were calculated for all the base learners (Fig. 2a, b). The ensemble consistently outperformed all base learners, on both datasets, achieving high ROC-AUC scores. For the bacteria-agnostic dataset, the ROC-AUC scores were 0.716 for the neural network, 0.736 for the logistic regression (LASSO), 0.719 for the random forest, 0.729 for the XGBoost and

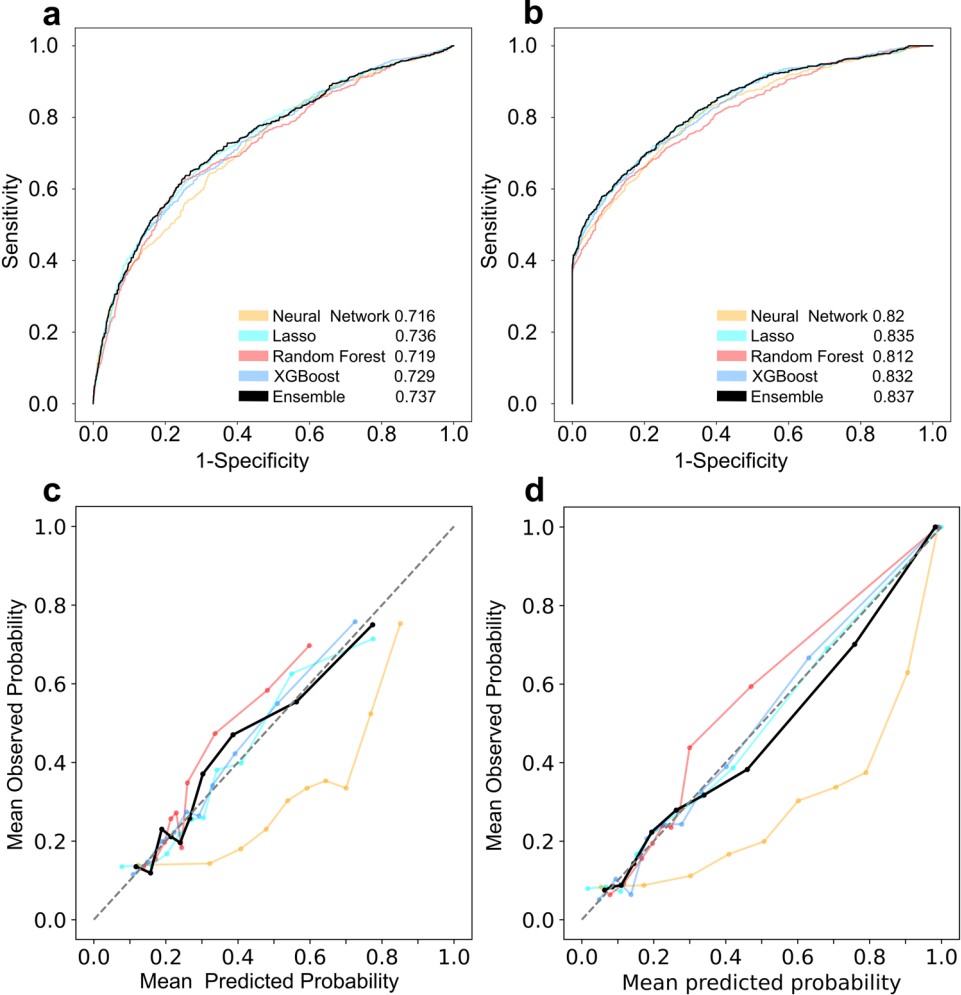

**Fig. 2 ROC curves and calibration plots for bacteria-agnostic and bacteria-gnostic datasets. a** ROC curve for bacteria-agnostic dataset, **b** ROC curve for bacteria-gnostic dataset **c** calibration plot for bacteria-agnostic dataset, **d** calibration plot for bacteria-gnostic dataset. ROC-AUC results of each model, on the test set, are presented within **a** and **b**. The colors represent different algorithms, where the black bold lines are the results of the ensemble model. Data points presented on the calibration plots are aggregated by deciles of predicted probability.

0.737 (95% CI 0.715-0.758) for the ensemble. On the bacteria-gnostic dataset the scores were 0.82 for the neural network, 0.835 for the LASSO, 0.812 for the random forest, 0.832 for the XGBoost and 0.837 (95% CI 0.821–0.854) for the ensemble. Furthermore, our ensemble models were well-calibrated (Fig. 2c, d).

In an effort to improve the ensemble's transparency and gain a better comprehension of the variables influencing its predictions, we used Kernel SHAP. This method estimates the contribution of each variable to the model's prediction by approximating their SHAP values[33]. These SHAP values allow us to understand the magnitude and direction of influence of variables, which implies variable importance (Fig. 3).

For the agnostic dataset, the five most influential variables in the bacteria agnostic dataset, as measured by the mean absolute SHAP values (Fig. 3a), were: previous resistance to ciprofloxacin in the past 60 days, whether the patient arrived from an institution, recent resistance to any antibiotic in same type of units (e.g., internal medicine or orthopedic units), previous resistance to ciprofloxacin during the previous 61–180 days, and recent resistance to any antibiotic in the hospital. Analogously, the five most influential variables in the bacteria gnostic dataset were (Fig. 3b): average resistance of the same bacterial species to any antibiotic in the past 30 days, across the hospital; the number of previous fluoroquinolone resistant infections the patient had in

the past 60 days; whether the bacterial species was *P. aeruginosa*; and the number of non-ciprofloxacin antibiotics that the same bacterial species had resistance to in the past 60 days, in the same patient. In both agnostic and gnostic settings, higher values of the influential variables consistently yielded positive influence on the ensemble's prediction, as can be seen by the swarm plots of the SHAP values (Fig. 3). This is simply the result of our coding of the binary variables (i.e., deciding which variable levels are set to zero or one) as risk factors.

Finally, we have performed a decision curve analysis (see Methods). Figure 4 shows that relying on predictions of our models can be at least as beneficial as assuming that every infection is resistant to ciprofloxacin, or assuming that every infection is sensitive to ciprofloxacin, for all cost-benefit ratios.

## Discussion

In this study, we developed two ensemble ML models to predict resistance to ciprofloxacin of hospitalized patients' infections. The first model was trained on the bacteria agnostic dataset, i.e., without any knowledge of the infecting bacterial species. This represents the most common situation before the start of antibiotic treatment. The second ensemble was trained on the bacteria gnostic dataset, i.e., with primary information of the infecting bacterial species.

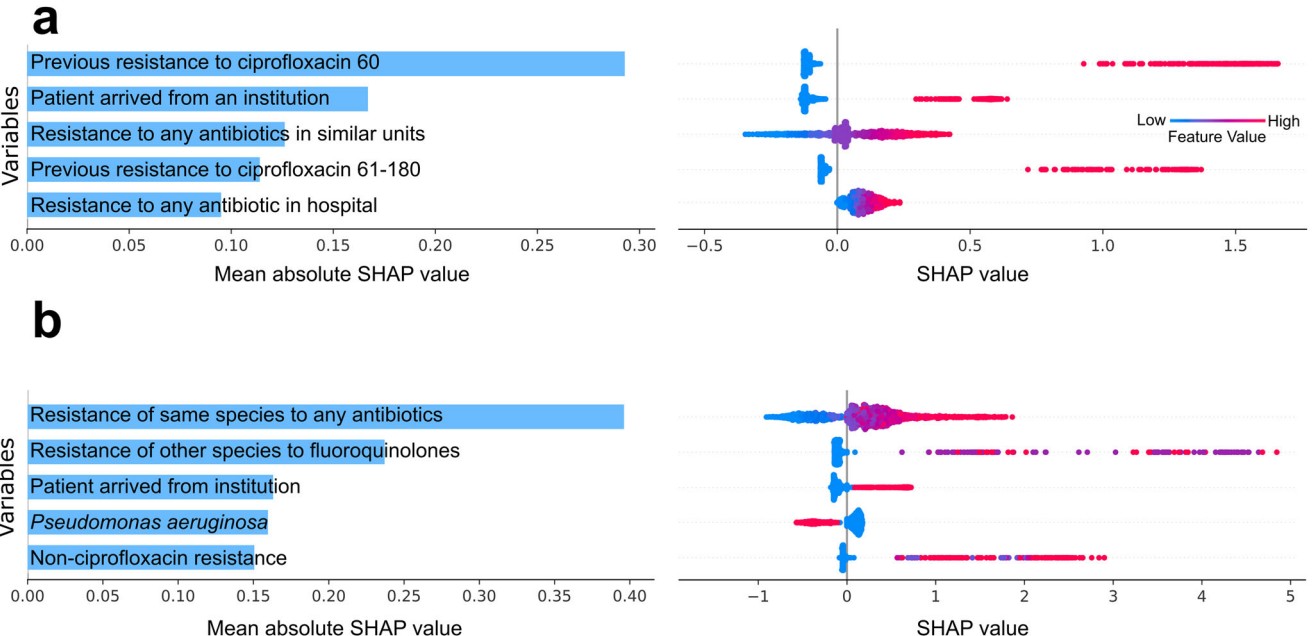

**Fig. 3 SHAP values of the ensemble model for the five most influential variables in the agnostic and gnostic datasets.** agnostic (**a**), gnostic (**b**). The absolute SHAP values are presented in the left column. A swarm plot is presented in the right column, wherein colors (from blue to red) correspond to variable values (from low to high), whereas the influence of those variables on the log-OR of predictions is given on the x-axis. Previous resistance to ciprofloxacin 60—whether the patient had a ciprofloxacin resistant infection in the past 60 days. Resistance to any antibiotic in similar units - past 30 days moving average of resistance to any antibiotic in the same type of units (orthopedic, gynecology etc.). Previous resistance to ciprofloxacin 61-180— whether the patient had a ciprofloxacin resistant infection in the 61-180 days prior to drawing the culture. Resistance to any antibiotics in hospital—past 30 days moving average of resistance to any antibiotic in the hospital. Resistance of the same bacterial species to any antibiotic - past 30 days moving average across the hospital. Resistance of other species to fluoroquinolones - the past 60 days, in the same patient. Non-ciprofloxacin resistance—number of non-ciprofloxacin antibiotics that the same bacterial species was resistant to in the last 60 days, in the same patient.

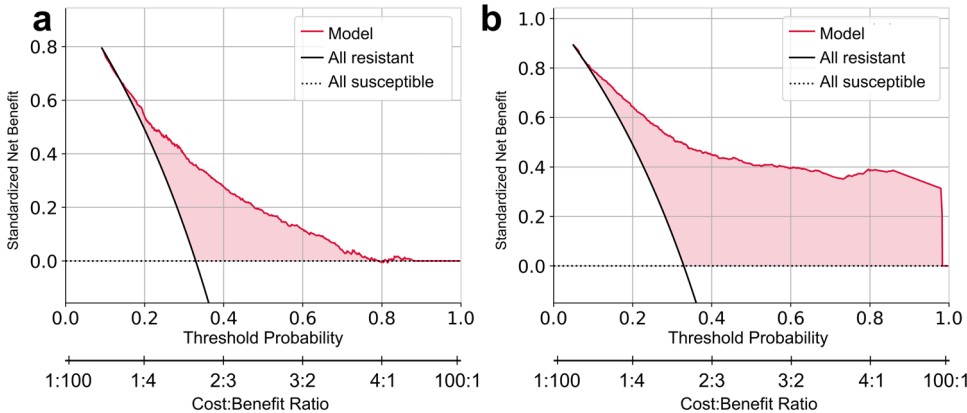

**Fig. 4 agnostic and gnostic decision curves.** agnostic (**a**), gnostic (**b**). The standardized net benefit is plotted against the threshold probability and cost-benefit ratio of deciding that an infection is resistant to ciprofloxacin. Curves of the benefit when assuming all infections are susceptible (dashed horizontal line), all infections are resistant (black curve), and relying on the ensemble model predictions (red curve) are plotted. Positive differences in standardized net benefit of the model predictions vs the all resistant and all susceptible curves are shaded in red.

Both models achieved high ROC-AUC metrics on an independent test set: 0.737 (95%CI 0.715–0.758) and 0.837 (95%CI 0.821–0.854) for the agnostic and gnostic datasets, respectively, and were well calibrated. Moreover, a decision curve analysis revealed that implementing our models can be beneficial in a wide range of cost-benefit considerations of withholding vs prescribing ciprofloxacin.

Our ML models include several innovative components in the field of AMR prediction. First, we use a super learner that is trained to effectively combine the outputs of several base learners. This increases our final ROC-AUC by up to 0.025 with respect to the base-learners. Second, we incorporate variables representing recent

and local resistant patterns within the hospital, in addition to a specific patient's EMR. Consequently, and despite the limited ability to compare such results between different settings, our models achieve high predictive abilities relative to previous studies[20,21]. Importantly, our models perform well on a very heterogeneous dataset, comprising various bacterial species, sample sources and multiple departments of the hospital. For example, Feretzakis et al.[21] predicted ciprofloxacin resistance using data from a single internal medicine department, conditioned on the sample's Gram stain result, and reached an ROC-AUC of 0.726[21]. Yelin et al.[20] predicted ciprofloxacin resistance only in outpatients, strictly using

urine samples, and limited to three bacterial species, reaching a ROC-AUC of 0.83[20]. Other studies either did not calculate ROC-AUC[23,44] or used cultures derived from a single sample source[22,23], a single bacterial species[44], or a single hospital unit[45].

An additional advantage of our ensemble modeling approach is built-in model calibration. Due to the logistic transformation the single-model outputs undergo, we are able to provide an output of well-calibrated probabilities of resistance. Prescribing antibiotics forces the clinician to make a compromise between patient's care and population-level consequences[46]. Hence, providing clinicians with unbiased probabilities of resistance can facilitate incorporation of other considerations into their decision. However, we note that continuous outputs from antibiotic prescription decision-support systems have been suggested to promote over-prescription of antibiotics, and hence decisions on output forms should be made with caution[47].

Our models' predictions were analyzed using SHAP values, which can aid in assessing the influence of different covariates on predictions when applying complex ML models[48]. We note that SHAP values contain inherent flaws[49] in approximating the impact of variables on predictions, and certainly do not aim to estimate causal effects. Despite these drawbacks, SHAP values can be useful for validating model outcomes against prior knowledge of risk factors and increase models' transparency. This can in turn facilitate increasing clinicians' trust in using ML decision support systems in their practice[50].

The results of our SHAP analyses are indeed consistent with the literature. Highly influential variables on the ensemble models' predictions were related to previous infections containing resistant bacteria, either to ciprofloxacin or other antibiotics. Previous resistance to ciprofloxacin is an obvious risk factor for current resistance[20,51,52]. However, the importance of previous resistance to other antibiotics may be explained by cross-resistance[53–55], or confounding by the patients' exposure to resistant bacteria or to antibiotics. Patients' origin (home, another hospital, nursing home, medical clinic, or other) had substantial influence on predictions and was also found to be an important variable by others[22,56]. This is a known risk factor, as antibiotics are administered more frequently in medical facilities and nursing homes, leading to high selection for resistance[57]. Local resistance frequencies, which we introduced into the data as moving averages of resistance frequencies, were also found to be highly influential on prediction. This is consistent with previous research and clinical use of local antibiograms, representing the susceptibility patterns of different bacteria[58]. Furthermore, our moving average of resistance frequencies is potentially more sensitive to resistance trends than yearly or monthly antibiograms. In the gnostic model, *P. aeruginosa* was selected as an influential variable. This stems from the binary encoding of the bacterial species, which defined the reference species as *E. coli*. Since *P. aeruginosa* was the second-most common bacterial species in the dataset, and was less resistant than *E. coli* (Supplementary Data 3), it was determined to be influential in reducing the predicted probability of a resistant infection. Finally, age was not deemed by our models as a highly important variable for ciprofloxacin resistance, in contrast to previous ML research 20 and classic retrospective studies[51,52]. This could potentially be attributed to the relatively old population in our study, especially when compared to studies on outpatients, which contain more heterogeneous cohorts.

Our study has several limitations. First, our dataset lacks relevant community-related patient information, such as antibiotic consumption in the community[57], and antibiotic consumption in the patients' surroundings, including neighborhoods[15] and households[59]. Our models can be easily extended to accommodate these covariates, which will likely further improve the models' predictive abilities. Second, our models are not necessarily immediately generalizable to other settings, or even the same setting, in different time periods. Variations in patients' demographics, antibiotic consumption, and the dynamic nature of AMR may lead to variation in risk factors over space and time[60,61]. For example, as we mention above, our data has under-representation of younger patients. This may be manifested in our model and needs to be taken in consideration when predicting resistance in young patients. Retraining of the models on site-specific data will likely be required to fine-tune predictions in different settings. However, the rates of ciprofloxacin resistance and patient covariates in our dataset are comparable with those of hospitalized patients in other developed countries[62]. We therefore expect a reasonable degree of consistency in our results, if our models would have been developed on a dataset from comparable settings.

## Conclusions

The models developed in this study represent a further step on the way to inclusion of ML decision support systems into clinical practice. Improvement of such models depends on advances in algorithm development, specific feature engineering, and the augmentation of the quantity and quality of EMR data. As we have shown, modern ML models can achieve high prediction while autonomously imparting high influence to risk factors that are known to be clinically relevant to AMR. Hopefully, future studies can further leverage the presented models and the vast EMR data available to improve prediction of AMR and consequently reduce antibiotic misuse.

## Data availability

Raw data is proprietary but can be made available upon reasonable request from the authors: The data pertains to the patient's electronic medical records. These are private and cannot be shared without approval from Meir Medical Center's IRB. Upon request, the authors and the individuals interested in accessing the data can write a formal request to the aforementioned IRB and seek its approval.

For the source data used to plot the resistance trends (Fig. 1), see Supplementary Data 4. For the source data used to plot the ROC curves, calibration and net benefit (Figs. 2 and 4) see Supplementary Data 5. For the source data used to plot SHAP (Fig. 3) see Supplementary Data 6.

## Code availability

The code is available at http://github.com/igormintz/cipro[63].

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

## Acknowledgements

This study was supported by the Israel Science Foundation (ISF 1286/21).

## Author contributions

I.M. and UO conceived the study; I.M. implemented the analysis; I.M., M.C., and U.O. interpreted the results; I.M. and U.O. wrote the initial draft of the manuscript; all authors revised and approved the final version of the manuscript.

## Competing interest

The authors declare no competing interests.
