## [Peer Review File · Communications Medicine]

Reviewers' comments:

Reviewer #1 (Remarks to the Author):

The authors have conducted a very nice study entitled "Prediction of Ciprofloxacin Resistance in Hospitalized Patients Using Machine Learning". This study could be of huge public benefit in terms of fighting AMR. The manuscript can be processed further for publication after addressing certain minor comments as below:

1. In the Abstract, write the full names of organisms.
2. The bacterial names in figure 1 should be italicized. And also, the second name should be in lowercase. E.g., *Staphylococcus aureus*
3. In table 1: add the spaces between organism names.
4. In table 1, is it possible to add the information regarding patients' stay in the hospital (in days) and if these were admitted in ICUs?
5. The authors have mentioned that the data was retracted from Meir hospital, but there is only 1 author affiliated with this hospital.
6. In the material and methods section, the authors can mention the details of bacterial culture testing, like what technique the hospital laboratory staff uses to identify and isolate the bacterial isolates, and how they determine the antibiotic susceptibility patterns of isolated organisms.
7. I can understand the importance of Ciprofloxacin, which is itself a very important antibiotic that can be used for both Gram-positive and Gram-negative bacteria, my question is why did the authors choose only Ciprofloxacin? Why not some other antibiotics too?

Reviewer #2 (Remarks to the Author):

Major comments: ^[1]_[SEP]

Introduction

Should include more information about the referenced studies about ciprofloxacin resistance prediction models - for example, what were the influential variables in these predictive models?

Data is missing -

<https://github.com/igormintz/cipro>

- Missing agnostics_22_06_22.csv and gnostics_22_06_22.csv files

- Not sure what these files include, but should include spreadsheets with all samples and their variables listed in Table S1 so that this study is reproducible

Figures

Figure 1 - unclear what the y-axis is as it is not labelled? Does 0.5 mean 50% of samples at a particular time were resistant? Would be helpful if the number of samples for training and test per species were indicated.

Methods

What does it mean that the additional features were "engineered" from the patients' EMR? Is this manual curation/extraction from patients' EMR or a script was written to parse the

EMRs?

Unclear whether what guideline standard (CLSI, EUCAST, etc) to establish susceptibility/resistance of bacterial isolates?

Very little resistance in *K. pneumoniae*, *P. aeruginosa*, and *S. aureus* - would that influence the prediction models?

Median is elderly population, do the authors believe that their evaluation metrics would if this model was applied to younger ages ?

The decision curve analysis portion is unclear to an audience who is unfamiliar, especially since it is used in the discussion to describe their influence on prescription decisions.

Minor comments:

- Last sentence of introduction is missing a period.
- Figure 1 - second part of species should be lowercase, e.g., *Escherichia coli* (not Coli)
- SAHP - misspelled (supposed to be SHAP)
- In decision curve analysis paragraph - refer to clinician as they instead of he/she
- Need to reference Python, Pandas, etc.
- Misspelled "gonstic"

Reviewer #3 (Remarks to the Author):

Mintz et al present a machine learning model designed to predict resistance to ciprofloxacin across a number of common Gram-negative bacteria and *S aureus*. The model was created using clinical and culture data from a single center in Israel and the authors reported ROC-AUCs, results of Shapley additive explanations analysis, and decision curve analysis.

The use of machine learning in predictive analytics has dramatically increased in recent years and there have been a number of recent studies utilizing both clinical and genomic data to predict antimicrobial resistance profiles. In this study, the authors utilize a previously described technique, but its application to predict ciprofloxacin resistance in an agnostic and gnostic fashion is novel.

However, there are two methodological concerns with the manuscript that should be either clarified or addressed.

1. The authors do not clearly state how the training and testing data set were selected. However, in Figure 1, they clearly demonstrate that all training data was captured from 2016-2019 and all testing data from 2019-2020. This time-based splitting can result in significant unidentified confounders with subsequent impact on the model. This effect can already be seen in the resistance trends for *K. pneumoniae* and *M. morganii* over time as depicted in Figure 1, as the resistance rate between the training set and the testing set appear to be significantly different. The authors should clarify why a time-based splitting was utilized or consider an alternative splitting strategy in order to make the model more generalizable.

2. The authors write that hyperparameter tuning was performed using 200 random searches with a five-fold, time-series cross-validation. However, it is not stated if the data used in parameter hypertuning were excluded from use in the training or testing sets; this should be clarified as resampling of the data used for hyperparameter tuning may artificially increase the ROC-AUC.

In general, this manuscript describes a generalizable technique to predict antimicrobial resistance based on clinical features, an important advancement in the management of multidrug resistant organisms. However, clarifications regarding methodology should be made in order to ensure the replicability of the authors' study and possibly improve the robustness of the model described.

12 January 2023

Dear Editor

Thank you for giving us the opportunity to submit a revision of our manuscript “Prediction of Ciprofloxacin Resistance in Hospitalized Patients Using Machine Learning” for consideration for publication in *Communications Medicine*. We are grateful to the reviewers for their thorough assessment of the manuscript and for their helpful suggestions for revision to improve it. Please see a point-by-point response (regular font) to the reviewers’ comments (**bold font**) below:

Reviewer #1 (Remarks to the Author):

The authors have conducted a very nice study entitled “Prediction of Ciprofloxacin Resistance in Hospitalized Patients Using Machine Learning”. This study could be of huge public benefit in terms of fighting AMR.

We thank the reviewer for this positive comment.

The manuscript can be processed further for publication after addressing certain minor comments as below:

1. In the Abstract, write the full names of organisms.

Fixed.

2. The bacterial names in figure 1 should be italicized. And also, the second name should be in lowercase. E.g., Staphylococcus aureus

Fixed.

3. In table 1: add the spaces between organism names.

Fixed.

4. In table 1, is it possible to add the information regarding patients' stay in the hospital (in days) and if these were admitted in ICUs?

Good Idea. We now added information about length of patients’ hospitalizations and the percentage of samples drawn in the ICU and in the ER to Table 1.

5. The authors have mentioned that the data was retracted from Meir hospital, but there is only 1 author affiliated with this hospital.

This is true. It is common and accepted in Israel to have collaborations between clinicians affiliated with a hospital and university researchers, using clinical data (which has been approved by the hospital, of course). This research is a product of such an ongoing collaboration between the clinician and university researcher.

6. In the material and methods section, the authors can mention the details of bacterial culture testing, like what technique the hospital laboratory staff uses to identify and isolate the bacterial isolates, and how they determine the antibiotic susceptibility patterns of isolated organisms.

Thanks for the comment. We added:

“For gram negative bacteria in urine or wound culture, VITEK 2 (bioMerieux, Durham, NC) was used. For all isolates from blood or for gram positive bacteria, in urine, wounds, or blood cultures, disk diffusion with CLSI breakpoints was used.”

7. I can understand the importance of Ciprofloxacin, which is itself a very important antibiotic that can be used for both Gram-positive and Gram-negative bacteria, my question is why did the authors choose only Ciprofloxacin? Why not some other antibiotics too?

Ciprofloxacin had the advantage of being both very frequently tested and having a high event rate. These provided us with sufficient data to develop relatively data-demanding and complex models, so it was chosen (i.e., even if a certain antibiotic is often tested but resistance to it is scarce, it would be hard to predict its resistance as the information content is low).

Reviewer #2 (Remarks to the Author):

Major comments:

Introduction

Should include more information about the referenced studies about ciprofloxacin resistance prediction models - for example, what were the influential variables in these predictive models?

Of the existing ML models that predict ciprofloxacin resistance, we could find only one that reported variable influence. We now refer to risk factors identified by it and by other, classical epidemiological approaches in the Discussion, in addition to various studies about known risk factors of antibiotic resistance:

“The results of our SHAP analyses are indeed consistent with the literature. Highly influential variables on the ensemble models’ predictions were related to previous infections containing resistant bacteria, either to ciprofloxacin or other antibiotics. Previous resistance to ciprofloxacin is an obvious risk factor for current resistance 20,51,52. However, the importance of previous resistance to other antibiotics may be explained by cross-resistance 53–55, or confounding by the patients’ exposure to resistant bacteria or to antibiotics. Patients’ origin (home, another hospital, nursing home, medical clinic, or other) had substantial influence on predictions and was also found to be an important variable by others 22,56. This is a known risk factor, as antibiotics are administered more frequently in medical facilities and nursing homes, leading to high selection for resistance 57. Another influential variable was sex. Associations between antibiotic resistance and sex have been observed repeatedly and may stem from differential antibiotic consumption patterns 21,22. Local resistance frequencies, which we introduced into the data as moving averages of resistance frequencies, were also found highly influential on prediction. This is consistent with previous research and clinical use of local antibiograms, representing the susceptibility patterns of different bacteria 58. Furthermore, our moving average of resistance frequencies is potentially more sensitive to resistance trends than yearly or monthly antibiograms. In the gnostic model, *P. aeruginosa* was selected as an influential variable. This stems from the binary encoding of the bacterial species, which defined the reference species as *E. coli*. Since *P. aeruginosa* was the second-most common bacterial species in the dataset, and was less resistant than *E. coli* (Table 1), it was determined to be influential in reducing the predicted probability of a resistant infection. Finally, age was not deemed by our models as a highly important variable for ciprofloxacin resistance, in contrast to previous ML research 20 and classic retrospective studies 51,52. This could potentially be attributed to the relatively old population in our study, especially when compared to studies on outpatients, which contain more heterogeneous cohorts.”

Data is missing -

<https://github.com/igormintz/cipro>

- Missing agnostics_22_06_22.csv and gnostics_22_06_22.csv files

As written in the “data availability” section, Data are proprietary but can be made available upon reasonable request from the authors. These are the original files used that are not shared in github. We now provide an explanation in the code as to the general format of such files.

- Not sure what these files include, but should include spreadsheets with all samples and their variables listed in Table S1 so that this study is reproducible

Same as previous comment.

Figures

Figure 1 - unclear what the y-axis is as it is not labelled? Does 0.5 mean 50% of samples at a particular time were resistant? Would be helpful if the number of samples for training and test per species were indicated.

We agree that this was unclear. We now added labels to the y-axis (fraction of resistant samples). We also added the number of samples for training and testing per species onto each plot.

Methods

What does it mean that the additional features were “engineered” from the patients’ EMR? Is this manual curation/extraction from patients’ EMR or a script was written to parse the EMRs?

Several scripts were written to create additional variables from the EMRs; e.g. average resistance in different points before the current test or categorization of resistance to different antibiotic classes. All the variables are described in SI table S1.

Unclear whether what guideline standard (CLSI, EUCAST, etc) to establish susceptibility/resistance of bacterial isolates?

Thanks for the comment. We added:

“For gram negative bacteria in urine or wound culture, VITEK 2 (bioMerieux, Durham, NC) was used. For all isolates from blood or for gram positive bacteria, in urine, wounds, or blood cultures, disk diffusion with CLSI breakpoints was used.”

Very little resistance in K. pneumoniae, P. aeruginosa, and S. aureus - would that influence the prediction models?

This indeed influences the models, as is seen in the differences in the agnostic and gnostic models. Knowing the bacterial species beforehand improved our results, partly because the average resistance rates between species differed.

Median is elderly population, do the authors believe that their evaluation metrics would if this model was applied to younger ages ?

Indeed, we expect our model, as almost any model, to perform worse on a population on which it has not been trained. This is why it is recommended to locally train such models per-setting of interest, to obtain population-specific parameters.

We now better explain this in the Discussion:

“Our models can be easily extended to accommodate these covariates, which will likely further improve the models’ predictive abilities. Second, our models are not necessarily immediately generalizable to other settings, or even the same setting, in different time periods. Variations in patients’ demographics, antibiotic consumption, and the dynamic nature of AMR may lead to variation in risk factors over space and time 60,61. For example, as we mention above, our data has under-representation of younger patients. This may be manifested in our model and needs to be taken in consideration when predicting resistance in young patients. Retraining of the models on site-specific data will likely be required to fine-tune predictions in different settings. However, the rates of ciprofloxacin resistance and patient covariates in our dataset are comparable with those of hospitalized patients in other developed countries 62. We therefore expect a reasonable degree of consistency in our results, if our models would have been developed on a dataset from comparable settings.”

The decision curve analysis portion is unclear to an audience who is unfamiliar, especially since it is used in the discussion to describe their influence on prescription decisions.

We appreciate the comment and recognize that this may be challenging for some readers. We have therefore made efforts to improve the clarity of this term in the methods section:

“A decision (also known as a utility) curve analysis, which is increasingly recognized as valuable in clinical predictive modeling 26, was performed using the predictions of our ensemble model on the test-set. A decision curve is a graphical representation of the trade-offs between the benefits and costs of a particular treatment or intervention, when administered according to a prognostic algorithm. It is used to evaluate the overall utility of the algorithm by considering both the magnitude of the benefits and costs of no-treatment and redundant treatment, and the likelihood of these results based on prevalence of the outcome and the algorithms’ prediction abilities.”

Minor comments:

- Last sentence of introduction is missing a period.

Fixed.

- Figure 1 - second part of species should be lowercase, e.g., Eschericia coli (not Coli)

Fixed.

- SAHP - misspelled (supposed to be SHAP)

Fixed.

- In decision curve analysis paragraph - refer to clinician as they instead of he/she

Fixed.

- Need to reference Python, Pandas, etc.

Fixed.

- Misspelled “gonstic”

Fixed.

Reviewer #3 (Remarks to the Author):

Mintz et al present a machine learning model designed to predict resistance to ciprofloxacin across a number of common Gram-negative bacteria and S aureus. The model was created using clinical and culture data from a single center in Israel and the authors reported ROC-AUCs, results of Shapley additive explanations analysis, and decision curve analysis.

The use of machine learning in predictive analytics has dramatically increased in recent years and there have been a number of recent studies utilizing both clinical and genomic data to predict antimicrobial resistance profiles. In this study, the authors utilize a previously described technique, but its application to predict ciprofloxacin resistance in an agnostic and gnostic fashion is novel.

However, there are two methodological concerns with the manuscript that should be either clarified or addressed.

1. The authors do not clearly state how the training and testing data set were selected. However, in Figure 1, they clearly demonstrate that all training data was captured from 2016-2019 and all testing data from 2019-2020. This time-based splitting can result in significant unidentified confounders with subsequent impact on the model. This effect can already be seen in the resistance trends for *K. pneumoniae* and *M. morganii* over time as depicted in Figure 1, as the resistance rate between the training set and the testing set appear to be significantly different. The authors should clarify why a time-based splitting was utilized or consider an alternative splitting strategy in order to make the model more generalizable.

We understand that we have not explained this sufficiently well: When working with time-dependent data, it is advisable to select a specific point in time to divide the data into training and testing sets, rather than randomly sampling the data, as external validation. This has several advantages. First, it is less prone to “data leakage”, where information from future observations informs past predictions (e.g. trends in resistance). This can artificially improve the model's performance. Second, this kind of time-based splitting emulates a real life-scenario where a hospital trains their model on past data and uses it for predictions from a certain point onwards, and is considered a form of external validation. The fact that the resistance trends may differ between train and test period may indeed detract from the model performance - as could happen in real-life ML applications - and we prefer this realistic approach. We now explicitly explain this in the Methods and provide appropriate references:

“The train-test split was performed based on calendar time, rather than randomly. This minimizes chances of “data-leakage”, where training on future observations holds information on past observations. Furthermore, such a split emulates a real-world scenario where the model can be trained up to a certain point and then used in the clinic from that point onwards, and is considered a form of external validation 24–27.”

2. The authors write that hyperparameter tuning was performed using 200 random searches with a five-fold, time-series cross-validation. However, it is not stated if the data used in parameter hypertuning were excluded from use in the training or testing sets; this should be clarified as resampling of the data used for hyperparameter tuning may artificially increase the ROC-AUC.

We agree that this was not clarified well-enough in the text. The training and test data are mutually exclusive, as best practice dictates. The hyper-parameter tuning

was not done on the test set. We also illustrate the process in SI figure S1. This now appears in the Methods:

“These datasets are mutually exclusive - all the presented results were obtained when training the models solely on the training set, and testing them on the independent test set.”

In general, this manuscript describes a generalizable technique to predict antimicrobial resistance based on clinical features, an important advancement in the management of multidrug resistant organisms. However, clarifications regarding methodology should be made in order to ensure the replicability of the authors' study and possibly improve the robustness of the model described.

Thank you for these kind words and we hope that the changes made have addressed these issues.

REVIEWERS' COMMENTS:

Reviewer #1 (Remarks to the Author):

The authors have made great efforts in revising the manuscript. I am okay with the revision. However, my concerns about figure 1 are still the same as I mentioned in the previous version. The authors should revise the bacterial names in figure 1, which should be italicized. And also, the second name should be in lowercase. E.g., *Staphylococcus aureus*.

Reviewer #2 (Remarks to the Author):

Thank you to the authors for addressing all of the reviewers' comments and concerns. I have no further comments and I believe this manuscript is now acceptable for publication.

Reviewer #3 (Remarks to the Author):

Obolski et al have provided the requested revisions for review. I appreciate the time they have taken to clarify the points I brought up previously regarding their methodology around training/testing set selection. The clarifications they provided improve the likelihood of reproducibility of their study and have addressed my comments satisfactorily.